# Advances in Therapy for Urothelial and Non-Urothelial Subtype Histologies of Advanced Bladder Cancer: From Etiology to Current Development

**DOI:** 10.3390/biomedicines13010086

**Published:** 2025-01-01

**Authors:** Whi-An Kwon, Ho Kyung Seo, Geehyun Song, Min-Kyung Lee, Weon Seo Park

**Affiliations:** 1Department of Urology, Myongji Hospital, Hanyang University College of Medicine, Goyang-si 10475, Republic of Korea; kein917@hanyang.ac.kr; 2Department of Urology, Center for Urologic Cancer, National Cancer Center, Goyang-si 10408, Republic of Korea; seohk@ncc.re.kr (H.K.S.); ghsong@ncc.re.kr (G.S.); 3Department of Internal Medicine, Myongji Hospital, Hanyang University College of Medicine, Goyang-si 10475, Republic of Korea; 4Department of Pathology, National Cancer Center, Goyang-si 10408, Republic of Korea

**Keywords:** antibody–drug conjugates, bladder cancer, chemotherapy, FGFR inhibitors, immunotherapy, non-urothelial cancer, subtype histologies, urothelial cancer

## Abstract

Urothelial carcinoma (UC) is the most common histological subtype of bladder tumors; however, bladder cancer represents a heterogeneous group of diseases with at least 40 distinct histological subtypes. Among these, the 2022 World Health Organization classification of urinary tract tumors identifies a range of less common subtypes of invasive UC, formerly known as variants, which are considered high-grade tumors, including squamous cell, small-cell, sarcomatoid urothelial, micropapillary, plasmacytoid, and urachal carcinomas, and adenocarcinoma. Their accurate histological diagnosis is critical for risk stratification and therapeutic decision-making, as most subtype histologies are associated with poorer outcomes than conventional UC. Despite the importance of a precise diagnosis, high-quality evidence on optimal treatments for subtype histologies remains limited owing to their rarity. In particular, neoadjuvant and adjuvant chemotherapy have not been well characterized, and prospective data are scarce. For advanced-stage diseases, clinical trial participation is strongly recommended to address the lack of robust evidence. Advances in molecular pathology and the development of targeted therapies and immunotherapies have reshaped our understanding and classification of bladder cancer subtypes, spurring efforts to identify predictive biomarkers to guide personalized treatment strategies. Nevertheless, the management of rare bladder cancer subgroups remains challenging because they are frequently excluded from clinical trials. For localized disease, curative options such as surgical resection or radiotherapy are available; however, treatment options become more limited in recurrence or metastasis, where systemic therapy is primarily used to control disease progression and palliate symptoms. Herein, we present recent advances in the management of urothelial and non-urothelial bladder cancer subtypes and also explore the current evidence guiding their treatment and emphasize the challenges and perspectives of future therapeutic strategies.

## 1. Introduction

Bladder cancer is a diverse group of cancers encompassing at least 40 distinct histological subtypes (formerly known as variants of urothelial carcinoma [UC]) that contribute to approximately 165,000 deaths globally each year [1]. The terminology “Urothelial Subtypes” refers to categories of bladder cancer that originate from urothelial cells but exhibit additional and distinct histological features beyond the conventional UC. These include UCs with mixed histological differentiation, such as squamous or glandular differentiation, or micropapillary UC. In contrast, “Non-Urothelial Subtypes” describe bladder cancers that arise from entirely different cellular lineages and not from urothelial cells. These subtypes include squamous cell carcinoma (SCC), adenocarcinoma, and small-cell carcinoma, which emerge through alternative pathways of differentiation and are distinct in both their histological appearance and cellular origin. The most common subtypes are UC, SCC, adenocarcinoma, and small-cell carcinoma, each exhibiting distinct clinical and biological characteristics [2]. Most bladder cancers are UC, formerly termed transitional cell carcinomas, whereas up to 10% are non-urothelial carcinomas [3].

In this review, we aim to provide a comprehensive synthesis of the current management landscape for advanced bladder cancer—encompassing both urothelial and non-urothelial subtypes—by integrating histopathological insights, evolving therapeutic strategies, and emerging molecular targets to better inform individualized treatment approaches.

For several decades, the therapeutic landscape for advanced bladder cancer remained relatively stagnant, with cytotoxic chemotherapy serving as the predominant treatment modality [4]. Commonly used agents included cisplatin and gemcitabine, which demonstrated efficacy in tumor reduction but were often associated with significant toxicity. This was because of their proven efficacy in inducing tumor shrinkage despite their significant toxicity and limited long-term benefits [5]. However, recent advancements have catalyzed a transformative era in bladder cancer management, driven by the advent of novel therapeutic agents, particularly in the domains of immunotherapy (e.g., pembrolizumab and atezolizumab) and molecularly targeted therapy (e.g., erdafitinib and enfortumab vedotin [EV]). These developments have renewed interest in the molecular and histopathological heterogeneity of bladder cancer, underscoring the need for personalized treatment paradigms that reflect the unique biological characteristics of these tumors. Specific molecular characteristics being targeted include FGFR3 mutations, HER2 amplification, and alterations in DNA damage repair genes, which provide potential pathways for precision therapies [6,7].

The 2022 World Health Organization (WHO) classification of urinary tract tumors delineates multiple subtypes, including both urothelial and non-urothelial carcinoma (Table 1). These subtypes may comprise the entire tumor or coexist with conventional UC components [1]. Although these subtypes are considerably less prevalent than pure UC, they are often associated with more aggressive clinical behaviors and poorer outcomes, necessitating meticulous management. Consistent with the WHO guidelines, all UC subtypes, as well as divergent differentiations, should be classified as high-grade malignancies, with a comprehensive documentation of all distinct histologic components to ensure accurate diagnosis and treatment planning. Classifying these subtypes as high-grade is clinically significant because it highlights the need for aggressive treatment and vigilant monitoring, given their association with a poor prognosis and increased risk of progression [8,9].

Despite the growing recognition of the importance of appropriately classifying and managing UC subtypes, most clinical trials have historically excluded patients with non-urothelial carcinomas or tumors exhibiting significantly divergent or subtype UC components [10]. Recently, efforts have been made to modify trial designs to include these subtypes, such as by creating dedicated cohorts for a subtype histology or incorporating adaptive trial designs to better evaluate responses to emerging therapies [11]. This exclusion is largely due to the rarity of these subtypes, their diverse biological behaviors, and the challenges in standardizing treatment protocols, which complicate the trial design and limit the ability to draw statistically significant conclusions. As a result, the management of these rare histological subtypes has largely depended on data extrapolated from other malignancies and limited clinical experience, often drawn from small case series that lack a robust understanding of their distinct biological features [2,12].

Many bladder cancers demonstrate some degree of divergent differentiation or contain at least one nonconventional UC histological subtype. These tumors are frequently managed in a manner similar to conventional UC [3]. A notable exception is small-cell carcinoma of the bladder (SCBCa), which, owing to its unique biological characteristics and aggressive behavior, is managed analogously to small-cell lung cancer (SCLC) irrespective of the proportion of tumors exhibiting this subtype [13,14].

This review provides a comprehensive overview of the current management strategies for advanced bladder cancer subtypes. It includes both pure forms and predominant forms (≥50% of the tumor consisting of a nonconventional UC component) of urothelial subtypes, as well as non-urothelial bladder cancer. We examine the role of histopathological evidence in guiding therapeutic approaches, including chemotherapy, immune checkpoint inhibitors, and targeted therapies. Furthermore, we offer an in-depth analysis of the pathology-driven management of subtypes of urothelial and non-urothelial carcinoma, highlighting the nuanced strategies required to optimize patient outcomes. Finally, we discuss emerging therapeutic modalities and innovations such as antibody–drug conjugates (ADCs) (e.g., EV) and FGFR inhibitors (e.g., erdafitinib), which have the potential to significantly broaden the current landscape of effective treatment options for these complex malignancies.

## 2. Etiology of Subtype Histologies in Urothelial Cancer

The subtype histology of UC represents a spectrum of distinct morphological subtypes that significantly diverge from conventional UC. These subtypes arise through a multifaceted interplay between genetic mutations, epigenetic modifications, tumor microenvironmental influences, and selective therapeutic pressures (Table 2). The convergence of these factors drives the development of the unique histological characteristics of UC [15,16,17].

Genetic alterations, such as mutations in key genes such as TP53, FGFR3, and RB1, lead to phenotypic diversity by disrupting cell cycle control, promoting unchecked cellular proliferation, and inducing resistance to apoptosis [18]. These mutations contribute significantly to the morphological and functional heterogeneity observed in various UC subtypes, highlighting the complexity of this malignancy [19].

Epigenetic changes, including DNA methylation and histone modifications, further influence gene expression in response to chronic inflammation and environmental stressors. For example, the hypermethylation of tumor suppressor genes results in their silencing, facilitating tumor growth and the emergence of subtype histologies [20,21]. Likewise, alterations in histone acetylation affect chromatin structure and gene transcription, thereby fostering diverse differentiation pathways. Importantly, these epigenetic changes are dynamic and can evolve due to tumor exposure to carcinogens, inflammatory cues, and therapeutic regimens. For instance, chemotherapy can induce DNA demethylation or modify histone patterns, leading to adaptive resistance and the emergence of aggressive histologic subtypes [22,23,24].

The tumor microenvironment, which consists of immune cells, fibroblasts, cytokines, and extracellular matrix components, plays a critical role in driving subtype differentiation via signaling interactions and hypoxic conditions. Key cytokines such as IL-6, TNF-α, and TGF-β are instrumental in this process. IL-6 promotes tumor growth and therapeutic resistance by enhancing inflammation and immune evasion. TNF-α drives inflammatory signaling that fosters cellular proliferation and differentiation, whereas TGF-β has a dual role—promoting epithelial–mesenchymal transition (EMT) and suppressing immune responses, thereby enhancing tumor invasiveness [25,26,27]. Chronic inflammatory states such as those induced by schistosomiasis or recurrent urinary tract infections further increase the likelihood of squamous differentiation (SqD) and alter the tumor’s histological characteristics [28,29,30]. Hypoxia within the tumor microenvironment activates hypoxia-inducible factors, which drive angiogenesis and metabolic adaptation and contribute to the development of aggressive histological subtypes. Additionally, interactions with cancer-associated fibroblasts and the remodeling of the extracellular matrix create a supportive niche that facilitates subtype differentiation and tumor progression [31,32].

The presence of cancer stem cells in UC further contributes to its histological diversity. These cells exhibit notable plasticity, which enables resistance to conventional treatments, including chemotherapy and radiation, by shifting their differentiation states. This plasticity complicates efforts to target these cells effectively using standard therapies, as their adaptive capacity allows them to evade treatment. This adaptability enables tumor cells to tolerate therapeutic stress, thereby contributing to tumor recurrence and progression [33,34]. Cancer stem cells express pluripotency markers such as OCT4, SOX2, and NANOG, maintaining self-renewal and adaptability, which are critical for the aggressiveness of the tumor [35,36].

Therapeutic interventions, including chemotherapy, radiation, and immunotherapy, are instrumental in shaping the landscape of subtype histology by applying selective pressure on tumor cells. Chemotherapy can result in sarcomatoid differentiation by selecting resistant clones that have undergone genetic or epigenetic adaptations, and radiation-induced DNA damage often promotes EMT, resulting in increased invasiveness and subtype morphologies [37,38,39]. ICIs, while potentially effective in some UC cases, may also influence the subtype histology by reshaping the immune landscape and possibly enabling immune escape mechanisms [40,41].

Specific genetic mutations are closely associated with distinct histological subtypes. FGFR3 mutations are frequently observed in the nested subtypes [42], whereas TP53 and RB1 mutations are prevalent in the sarcomatoid and neuroendocrine subtypes [43,44]. The identification of these mutations through molecular profiling is crucial for prognostication and for guiding personalized therapeutic strategies.

## 3. Clinical Development for Management of Urothelial Subtype and Non-Urothelial Bladder Cancer

### 3.1. Urothelial Subtype Bladder Cancer

Urothelial subtype bladder cancers represent a distinct subset of rare malignancies characterized by unique clinical and biological phenotypes, including increased invasiveness, higher rates of metastatic spread, and distinct histological features. A defining feature of these subtypes is their disproportionately high morbidity, relative to their low incidence. Moreover, the scarcity of prospective clinical trial data poses significant challenges for clinicians in determining optimal therapeutic strategies, particularly in metastatic settings, owing to difficulties in patient recruitment due to the rarity of these subtypes [45].

The most prevalent of these atypical histological subtypes include micropapillary urothelial carcinoma (MPUC), sarcomatoid urothelial carcinoma (SUC, also known as carcinosarcoma), and plasmacytoid urothelial carcinoma (PUC). Each subtype exhibits distinct biological behaviors and clinical trajectories, necessitating sophisticated and tailored management approaches that diverge from those used for conventional UCs [10].

#### 3.1.1. Micropapillary Urothelial Cancer

MPUC is a rare and highly aggressive subtype of UC, accounting for approximately 0.01–2.2% of urothelial tumors [46]. Histologically, MPUC is characterized by tight infiltrative clusters of micropapillary aggregates. These aggregates are frequently associated with vascular and lymphatic invasion, contributing to significant metastatic potential and adverse clinical outcomes (Figure 1A) [10]. First described in 1994 at the MD Anderson Cancer Center, MPUC demonstrates morphological similarities to ovarian papillary serous tumors, which is significant because of their shared aggressive behavior and poor prognosis, suggesting potential parallels in treatment approaches [47]. It typically presents at an advanced stage and is characterized by small nests of tumor cells within the lacunae, often lacking vascular cores, and it is accompanied by extensive lymphovascular invasion. Despite its aggressive nature, definitive histopathological criteria for MPUC remain controversial, further complicating its diagnosis and subsequent management [48].

The management of MPUC has been challenging, largely because of its rarity and the limited availability of high-quality evidence to guide treatment decisions, particularly regarding the role of chemotherapy [49]. Retrospective analyses suggest that micropapillary non-muscle invasive bladder cancer carries a substantial risk of progression to muscle-invasive disease, prompting many experts to advocate upfront radical cystectomy (RC) over intravesical therapies such as Bacillus Calmette-Guerin (BCG) [50]. Nevertheless, conflicting data persist, with some studies indicating no significant difference in survival outcomes between immediate RC and intravesical BCG [51,52].

The efficacy of neoadjuvant chemotherapy (NAC) in the treatment of localized MPUC remains controversial. A retrospective analysis of 869 patients from the US National Cancer Database demonstrated that NAC combined with RC was associated with no survival advantage compared to RC alone [46]. In contrast, a separate investigation conducted at the Memorial Sloan Kettering Cancer Center (MSKCC) reported that NAC significantly increased the rate of pathological downstaging to pT0 (13% vs. 45%; *p* = 0.049), suggesting a potential benefit in selected patient populations [53,54]. However, the absence of randomized controlled trials (RCTs) and the reliance on retrospective data introduce significant uncertainty, complicating the development of standardized treatment recommendations.

Recent molecular studies have provided critical insights into the pathobiology of MPUC, including HER2 overexpression [55] and the identification of luminal and p53-like subtypes [56], thereby elucidating potential therapeutic targets. HER2 overexpression suggests a rationale for investigating HER2-directed therapies such as trastuzumab, pertuzumab, and lapatinib in clinical trials, which have demonstrated efficacy in HER2-positive breast and gastric cancers [57,58]. Furthermore, the identification of luminal and p53-like subtypes implies that immunotherapy or subtype-specific targeted approaches, such as FGFR inhibitors for luminal subtypes [59] or PARP inhibitors for p53-like subtypes [60], may offer clinical benefits. Comprehensive mRNA expression profiling has demonstrated that most MPUC tumors correspond to the luminal subtype. Specifically, these tumors are stratified within The Cancer Genome Atlas clusters I (urothelial-like) and II (p53-like, infiltrated) [61].

There is a pressing need for studies evaluating the efficacy of NAC, identifying predictive biomarkers for treatment selection, optimizing personalized therapeutic strategies, and fostering international collaboration to mitigate the challenges of patient recruitment because of the rarity of MPUC [62]. Additionally, although data on the use of immunotherapy, targeted therapy, and ADCs in MPUC are limited, the preliminary results are encouraging. For example, an overall response rate of 28% was reported in a retrospective cohort of 25 patients with micropapillary bladder cancer treated with ICIs [63]. Similarly, in a separate retrospective cohort of 41 patients with micropapillary components treated with EV, the overall response rate was 35%. Specifically, among patients with micropapillary-predominant tumors, one out of five responded to therapy, although no cases of pure micropapillary carcinoma were included in this cohort [64]. Given the rarity of MPUC, the available evidence is predominantly derived from retrospective studies, which are inherently limited by potential biases.

There is an urgent need for prospective clinical trials specifically designed to evaluate this subtype, with a focus on the efficacy of neoadjuvant and adjuvant therapies, as well as the exploration of novel targeted approaches. Until more definitive data become available, treatment strategies should be individualized based on the disease stage, patient comorbidities, and molecular tumor characteristics, with a multidisciplinary care team to ensure a comprehensive and nuanced approach.

#### 3.1.2. Plasmacytoid Urothelial Cancer

PUC is a rare, aggressive histological subtype of UC, accounting for 1–3% of cases. First described in 1991, PUC is characterized by cells resembling plasma cells with a high proliferative index, as evidenced by frequent Ki-67 protein expression [65,66]. PUC tends to involve the bladder wall extensively and frequently invade adjacent organs, such as the ureters, rectum, prostate, and perivesical soft tissues. Its spread along the pelvic fascial planes complicates surgical management, often resulting in positive resection margins and surgical upstaging during cystectomy [67].

Because of its rarity, the limited data on PUC present significant challenges for clinical decision-making and the establishment of standardized treatment protocols, with these often relying on case reports and small institutional studies. Historically, PUC has been occasionally misdiagnosed as plasma cell myeloma upon bone metastasis, underscoring the diagnostic challenges arising from its morphological resemblance to plasma cells (Figure 1B). PUC diffusely infiltrates the bladder wall, exhibiting a pattern similar to the linitis plastica phenotype observed in gastric and lobular breast carcinomas. This phenotype is associated with an extensive local spread and poor prognosis, which complicates clinical management and surgical planning. PUC typically presents at the muscle-invasive and advanced stages, with a pronounced tendency for early peritoneal dissemination along the fascial planes, contributing to symptoms of gastrointestinal and urinary obstruction [68,69].

Clinically, PUC is associated with advanced disease at presentation and a poor prognosis. Patients with PUC frequently exhibit higher-stage disease, positive lymph node involvement, and positive soft tissue surgical margins compared with those with conventional UC. The median overall survival is 19 months, which reflects the aggressive nature of PUC. Management of PUC often involves aggressive treatment strategies. Although NAC may achieve pathological downstaging, response rates are inconsistent and relapse is common. Upfront RC is often performed; however, the high incidence of non-organ-confined disease necessitates the strong consideration of adjuvant cisplatin-based chemotherapy [70].

Despite demonstrating some degree of chemosensitivity, the responses of PUC to NAC are typically limited and short-lived, likely due to intrinsic resistance mechanisms such as alterations in DNA damage repair pathways and high rates of cadherin-1 (CDH1) mutations [68]. Pathological downstaging after NAC is consistently lower in patients with PUC than in those with conventional UC, with studies indicating low downstaging rates and more frequent cases of unresectable disease [62]. Compared with UC, PUC exhibits significantly poorer outcomes following NAC, including reduced downstaging rates and shorter relapse-free and overall survival (OS) [71]. These findings highlight the urgent need for novel therapeutic approaches to improve the outcomes of patients with PUC.

Genetically, PUC frequently harbors mutations or the promoter hypermethylation of CDH1, leading to loss of E-cadherin function. This loss contributes to its discohesive morphology and diffuse infiltrative growth, complicating accurate staging and leading to intraoperative challenges in delineating tumor boundaries [72]. Studies have reported that PUC results in increased rates of positive surgical margins and ureteric tumor spread compared with other UC subtypes, necessitating more extensive surgical approaches and contributing to adverse postoperative outcomes, including higher recurrence rates and complications [73].

PUC frequently exhibits alterations in the p53, RB, and DNA damage repair pathways, in addition to a high tumor mutation burden and immune-infiltrated phenotype [73], suggesting its potential responsiveness to ICIs such as pembrolizumab and atezolizumab. Clinical studies have reported response rates to ICIs ranging from 17% to 32%, with a median response duration of up to 17 months [63,74]. Additionally, ADCs targeting nectin-4 and trop-2 have shown promise for the treatment of PUC. A high expression of nectin-4 and trop-2 has been documented in PUC specimens [75], and the UNITE retrospective dataset reported no response to EV in pure UC subtype histology cases. However, among 23 mixed UC cases with a plasmacytoid component, the response rates ranged from 53% to 64%, depending on the plasmacytoid percentage [64].

PUC is a rare and highly aggressive subtype of UC with distinct biological characteristics and a poor prognosis. Its discohesive morphology and diffuse infiltration, driven by CDH1 alterations, complicate surgical management. Future research must prioritize prospective clinical trials to evaluate the efficacy of NAC, ICIs, and ADCs in PUC treatment, with consideration of trial designs focusing on molecular stratification, optimal sequencing of therapies, and endpoints such as OS, progression-free survival, and quality of life. Clinicians should emphasize individualized treatment strategies incorporating molecular profiling and multidisciplinary approaches to optimize patient outcomes.

#### 3.1.3. Sarcomatoid Urothelial Cancer

SUC is a rare and highly aggressive subtype of UC, accounting for approximately 0.1–0.3% of all bladder tumors. This malignancy is characterized by high-grade sarcomatous differentiation, with approximately 50% of patients presenting with metastatic disease at the time of diagnosis, resulting in a poor prognosis. The development of SUC is often linked to prior exposure to DNA-damaging agents such as cyclophosphamide or radiotherapy, which induce mutational events such as TP53 alterations [76]. Histologically, SUC consists of malignant spindle cells with a mesenchymal phenotype and may differentiate into diverse mesenchymal lineages, including osteosarcoma, rhabdomyosarcoma, or chondrosarcoma (Figure 1C). The WHO classifies SUC as a high-grade malignancy owing to its mixed epithelial and mesenchymal components, complicating its differentiation from pure sarcomas. Immunohistochemical markers, such as cytokeratin and GATA3, are essential for diagnostic confirmation, and the extent of the sarcomatoid component (>50%) is a critical prognostic factor, warranting specific reporting in pathological evaluations [77,78].

Therapeutic strategies for SUC remain poorly defined and rely largely on evidence extrapolated from limited case series or analogous malignancies. In cases of localized SUC, RC is recommended, particularly for patients with non-muscle-invasive disease who are BCG-naïve, as these patients are unlikely to derive substantial benefits from intravesical BCG therapy. For muscle-invasive, non-metastatic SUC, RC is the mainstay of treatment, with retrospective studies demonstrating an OS advantage, including increased 5-year survival rates [79]. Recent studies have increasingly focused on the use of NAC to treat muscle-invasive diseases. While earlier retrospective analyses suggested that patients with SUC who received NAC experienced only a minimal and statistically insignificant improvement in survival, a more recent meta-analysis demonstrated a significant survival benefit for those treated with NAC compared with those who did not receive NAC. However, response rates may vary across different histological SUC subtypes, with some subtypes showing greater sensitivity to NAC than others [80,81]. This variability highlights the importance of individualized treatment planning.

For advanced or metastatic SUC, treatment options are limited and are primarily derived from broader UC studies. Regimens such as gemcitabine and cisplatin (GC) have shown efficacy, including reports of complete responses in small case studies [82]. Given the high levels of PD-L1 expression observed in SUC, ICIs such as anti-PD-1/PD-L1 agents have emerged as promising therapeutic options, with retrospective studies reporting response rates of 35–40% [63].

At the molecular level, SUC frequently harbors mutations in TP53, RB1, and PIK3CA. These mutations have significant implications for treatment and prognosis. TP53 mutations are often associated with a poor response to standard chemotherapeutic regimens, whereas RB1 loss has been linked to increased sensitivity to cell cycle inhibitors. PIK3CA mutations, on the other hand, suggest a potential responsiveness to targeted therapies involving PI3K inhibitors, highlighting the importance of molecular profiling in guiding personalized treatment strategies [78,83]. Molecular profiling has identified two primary subtypes, basal and double negative, with the basal subtype potentially showing greater sensitivity to targeted therapies such as EGFR or FGFR inhibitors [83]. Compared to conventional UC, SUC demonstrates an elevated expression of immune-related genes, including PD-L1, with approximately 50% of the cases showing significant overexpression [83]. The aggressive nature of SUC is partly attributable to EMT, which enhances its metastatic capacity [84].

Recent attention has been directed toward the role of ADCs in SUC treatment. Recent advancements in ADC technology, including optimized linker chemistry and novel payloads, aim to enhance therapeutic outcomes and broaden their applicability, particularly in challenging tumor types such as SUC. ADCs selectively deliver cytotoxic agents to cancer cells by targeting specific antigens, thereby minimizing off-target effects and potentially reducing systemic toxicity. However, the efficacy of ADCs may be constrained by heterogeneous antigen expression across tumor populations. EV, an ADC targeting nectin-4, is currently employed in combination with pembrolizumab as a first-line treatment for metastatic UC [85]. However, SUC is characterized by a relatively low nectin-4 expression, which may diminish the therapeutic effectiveness of EV [86]. Small retrospective series have suggested variability in the response to EV based on the proportion of sarcomatoid components, with better responses noted in tumors with a lower sarcomatoid content [64]. Ongoing clinical trials are evaluating the efficacy of EV in combination with pembrolizumab for SUC and other non-conventional histologies (ClinicalTrials.gov identifier: NCT05756569).

SUC remains challenging because of its aggressive nature, heterogeneous histology, and limited treatment options. Advances in molecular profiling and emerging therapeutic strategies, such as ICIs and ADCs, have provided hope for improved outcomes. However, a more profound understanding of the molecular underpinnings of SUC and the establishment of evidence-based clinical guidelines are urgently required to enhance patient management and prognosis.

### 3.2. Non-Urothelial Subtype Bladder Cancer

#### 3.2.1. Squamous Cell Bladder Cancer

Bladder SCC is characterized by cohesive tumor cells that exhibit keratinization and intercellular bridges, often originating from keratinizing squamous metaplasia (Figure 1D) [87]. While uncommon in Western countries (2–3%) [88], its prevalence is higher in African regions where *Schistosoma haematobium* infection is endemic. Additional risk factors include chronic bladder irritation caused by recurrent infections, urolithiasis, and prolonged catheter use [28]. A large-scale U.S. study reported a 2.4% incidence of SCC in more than 160,000 bladder cancer patients [89].

The prognosis of localized bladder SCC is typically poor, driven by the high propensity for local invasion observed in over 80% of the cases. In contrast to UC, which often metastasizes to distant sites, SCC tends to remain localized, leading to aggressive local disease and unfavorable outcomes [90]. Studies evaluating survival outcomes in patients undergoing RC and pelvic lymph node dissection found no significant differences in OS between pure SCC and UC, after controlling for lymph node involvement and soft tissue margins [91].

The primary treatment modality for localized pure SCC is surgical resection, with chemoradiotherapy considered in cases where surgery is contraindicated or when the patient is deemed unsuitable for surgery. According to the National Comprehensive Cancer Network (NCCN) guidelines, surgical intervention is preferred, with postoperative radiation therapy considered in cases with positive surgical margins. The role of NAC or adjuvant chemotherapy and/or radiation remain unclear because of the rarity of SCC [92], although some retrospective analyses suggest that adjuvant radiation therapy may reduce pelvic recurrence [93].

Retrospective analyses of different treatment strategies for SCC have produced mixed results, with some studies showing poor outcomes due to rapid disease progression or treatment-related toxicity, whereas others have indicated improved survival following surgical intervention [94].

Despite the scarcity of prospective data, largely owing to the rarity of SCC and challenges in enrolling sufficient patient numbers for rigorous trials, platinum-based chemotherapy has demonstrated efficacy in SCC [95]. This lack of robust data complicates the development of standardized treatment recommendations. The role of chemotherapy in SCC remains controversial, given the paucity of robust evidence and the inherently aggressive nature of the disease. Historically, platinum-based regimens have been considered largely ineffective for SCC, although this may reflect the intrinsic aggressiveness of the disease rather than absolute chemoresistance [96]. However, recent data have suggested that a subset of patients may benefit from chemotherapy. Specifically, an ORR of 35% was observed in patients treated with ifosfamide, paclitaxel, and cisplatin [97]. A large study involving over 400 patients with advanced bladder cancer treated with platinum-based chemotherapy found no significant differences in response rates between UC, pure SCC, and SCC subtypes of UC (44%, 27%, and 34%, respectively; *p* = 0.21) [98]. Moreover, a phase II study evaluating patients with bilharzia-associated bladder cancer reported comparable response rates to gemcitabine and cisplatin between patients with UC and those with SCC (60% vs. 50%) [95]. The efficacy of taxanes, such as paclitaxel, in SCCs of other anatomical sites (e.g., lung, head, and neck) has also prompted the exploration of bladder SCC. However, the combination of paclitaxel, ifosfamide, and cisplatin has demonstrated substantial toxicity, limiting its use in metastatic settings [97].

The molecular characterization of pure bladder SCC remains in its nascent stages owing to limited sample availability, genetic heterogeneity, and the rarity of the disease. Preliminary data indicate that genetic alterations in SCC differ significantly from those in UC, with a notable tendency for loss of chromosome arm 3p [99]. The gene expression profiles of SCC and UC with SqD also exhibit substantial divergence, suggesting that epigenetic mechanisms and non-coding RNAs are pivotal in determining the tumor phenotype. Specifically, the hypermethylation of tumor suppressor genes and alterations in histone modifications have been implicated in SCC. Additionally, non-coding RNAs, such as miR-21 and lncRNA HOTAIR, have been found to play roles in regulating SqD and tumor progression. Genetic drivers of SqD common across SCCs at various anatomical sites, such as NOTCH1, TP63, and SOX2, may provide a basis for the development of targeted therapies specific to bladder SCC [100].

Emerging therapeutic modalities, including ICIs and ADCs, have shown promise in treating non-urothelial carcinoma (non-UC) bladder cancers, particularly SCC and other histologic subtypes such as the micropapillary and plasmacytoid urothelial subtypes. A phase I study involving cabozantinib and nivolumab demonstrated favorable responses in two out of two SCC patients treated in an advanced setting, with one experiencing a complete radiographic response and the other experiencing a partial response [101]. Furthermore, PD-1/PD-L1 inhibitors showed a response rate of 28% in 50 patients with UC with SqD, comparable to that of pure UC in terms of progression-free and OS [63].

Recently, phase II trials have provided significant insights. For instance, the PURE-01 phase II trial involving pembrolizumab in a neoadjuvant setting demonstrated encouraging results, achieving tumor downstaging in 86% (6 of 7) of tumors with a predominant squamous component (>50%) [102].

EV, an ADC targeting nectin-4, has demonstrated efficacy in patients with a non-UC histology, including those with SqD. In a retrospective study, high nectin-4 expression was observed in the squamous component of UC in 7 of 10 patients [86]. The UNITE study reported a 50% response rate to EV monotherapy in patients with UC with SqD; however, no response was observed in patients with pure SCC [103]. An ongoing phase II trial is evaluating the combination of EV and pembrolizumab for the treatment of SCC and other non-UC histologies (ClinicalTrials.gov identifier: NCT05756569). Sacituzumab govitecan (SG), an ADC targeting Trop-2, has also demonstrated efficacy against non-UC bladder cancer. In a retrospective analysis, patients with any form of SqD had an ORR of 21%, whereas those with pure SCC had an ORR of 24% [104]. An ongoing phase II trial is assessing the effects of SG prior to RC in muscle-invasive non-UC bladder tumors (ClinicalTrials.gov identifier: NCT05581589).

The management of pure SCC of the bladder remains challenging owing to limited prospective evidence and the aggressive nature of the disease. Surgical resection continues to be the cornerstone treatment for localized disease, with adjuvant radiation potentially reducing locoregional recurrence. Although chemotherapy has historically been viewed as less effective, recent studies have suggested that some patients may benefit from platinum-based regimens. Immunotherapy and targeted approaches, particularly involving ADCs, represent promising areas for future investigation given the shared molecular characteristics between bladder SCC and SCCs at other anatomical sites.

#### 3.2.2. Neuroendocrine Bladder Cancer

Neuroendocrine carcinoma of the bladder is a rare and highly aggressive subset of bladder malignancies that accounts for approximately 2% of all bladder cancer cases. Among these, SCBCa is the most prevalent subtype, comprising approximately 0.5–1.2% of bladder cancers, and it represents the predominant form of extrapulmonary neuroendocrine carcinoma within the genitourinary system [3]. Owing to its rarity, management strategies for SCBCa are primarily derived from retrospective studies and case reports, because RCTs are largely unavailable.

Key risk factors for SCBCa include tobacco use, male sex, and age > 60 years. Genetic predispositions such as TP53 and RB1 mutations, along with environmental exposure, are also believed to contribute to disease risk [105]. Histologically, SCBCa exhibits characteristics similar to SCLC. It is characterized by small, round, hyperchromatic cells with scant cytoplasm that stain blue with hematoxylin and eosin (Figure 1-E). These cells are immunoreactive for neuroendocrine markers, including chromogranin, synaptophysin, CD56, and neuron-specific enolase. Furthermore, 38–70% of SCBCa cases also contain components of non-small-cell carcinoma, suggesting a shared clonal origin from multipotent cancer stem cells capable of divergent differentiation [106].

SCBCa is characterized by a high somatic mutational burden, predominantly driven by APOBEC-mediated mutations that contribute to genomic instability and a highly aggressive phenotype. These mutations lead to frequent alterations in TP53, RB1, and TERT promoters, resulting in deregulated cell proliferation and enhanced tumor growth. The loss of TP53 and RB1 contributes to unchecked cellular proliferation and genomic instability, whereas TERT promoter mutations enhance telomerase activity and facilitate sustained tumor growth. However, the fundamental biological mechanisms underlying SCBCa remain poorly characterized [107].

There is a central debate regarding the optimal management of SCBCa. One perspective suggests that it should be approached similarly to UC, because of its bladder origin and potential responsiveness to NAC. Alternatively, it may be treated akin to other small-cell carcinomas, such as SCLC, given their shared histopathological characteristics and responsiveness to platinum-based regimens. Comprehensive staging, including brain imaging, is crucial because over 60% of patients present with metastatic disease at diagnosis [108].

SCBCa follows an aggressive clinical course, and patients diagnosed with advanced disease have a median OS of approximately 12 months. Even in the context of localized disease, the median OS is <20 months. Multimodal treatment strategies, including NAC followed by RC or chemoradiotherapy, have demonstrated improved outcomes compared to monotherapy for localized SCBCa [108].

Surveillance, Epidemiology, and End Results (SEER) analyses have demonstrated an increasing trend in the use of NAC followed by surgery in the United States between 2001 and 2016 [109]. In a prospective phase II study, patients with organ-confined SCBCa who were treated with alternating ifosfamide–doxorubicin and etoposide–cisplatin achieved pathological downstaging to ≤ypT1 in 78% of cases, with a median OS of 58 months, underscoring the importance of systemic therapy in localized SCBCa [14].

The management of metastatic SCBCa is particularly challenging, with outcomes that are significantly worse than those of UC. Although metastatic SCBCa initially responds well to chemotherapy, the duration of the response is typically short due to mechanisms such as drug resistance, tumor heterogeneity, and aggressive tumor biology, with a median OS ranging from 10.3 to 13.3 months [110]. Platinum-based chemotherapy with etoposide remains the standard first-line treatment for metastatic SCBCa, largely extrapolated from SCLC treatment paradigms [108].

The role of immunotherapy in SCBCa remains under investigation, and early-phase trials have yielded mixed results. Although some studies have demonstrated promising effects, others have failed to show significant benefits. For example, a phase I trial combining pembrolizumab with etoposide reported a response rate of 43%, whereas a phase II trial involving durvalumab and tremelimumab reported no significant response [3]. A phase II/III trial (NCT05058651) is currently evaluating the efficacy of platinum-based induction chemotherapy and etoposide, with or without atezolizumab, followed by atezolizumab maintenance in patients with extrapulmonary small-cell carcinoma. Given the limited data available on subsequent lines of treatment, participation in clinical trials is crucial. Emerging targeted therapies such as lurbinectedin and SG have demonstrated clinical activity in small studies, although the efficacy of dual immune checkpoint blockade has been inconsistent.

Nectin-4, a target for EV, is expressed at low levels in SCBCa, thereby limiting EV’s therapeutic potential in this setting [64]. Another promising target is delta-like ligand 3 (DLL3), which is expressed in various neuroendocrine malignancies including SCBCa. DLL3-targeted therapies, such as the tri-specific T-cell-activating construct HPN328, are being evaluated in early-phase clinical trials and have shown encouraging clinical activity in metastatic SCBCa [111].

SCBCa is a rare yet highly aggressive subtype of bladder cancer characterized by a high metastatic potential and poor OS outcomes. Multimodal treatment strategies incorporating systemic therapy are critical for optimizing outcomes in localized disease; however, the management of metastatic SCBCa remains a formidable challenge. Further research is urgently needed to elucidate the underlying biology of SCBCa, focusing on genomic studies, novel immunotherapeutic targets, and the exploration of combination therapies to develop more effective treatments.

#### 3.2.3. Adenocarcinoma: Non-Urachal and Urachal Subtype

Bladder adenocarcinoma (BAC) represents a rare subtype of bladder cancer, accounting for approximately 0.8% of all bladder malignancies, based on an analysis of over 200,000 patients included in the 2004–2016 SEER database [112]. These tumors present in various morphological patterns, including papillary, nodular, flat, or ulcerative forms, and typically localize to the trigone and posterior bladder wall [113]. Histologically, BAC can be subdivided into enteric, mucinous, signet ring cell, and mixed subtypes [1]. Microscopically, these tumors often exhibit a well to moderately differentiated glandular morphology, closely resembling colorectal adenocarcinoma, with or without extensive mucin production (Figure 1F).

BAC can be further classified into urachal adenocarcinoma (UrCA) and non-urachal adenocarcinoma. Non-urachal adenocarcinoma is the more prevalent form and is generally associated with a worse prognosis. UrCA accounts for approximately 10% of BAC cases and typically involves the dome and anterior aspect of the bladder wall [113]. This localization is likely attributable to its embryological origin in the urachus, which is a remnant of the allantois that connects the bladder to the umbilicus. Compared to non-urachal adenocarcinoma, UrCA tends to present at a later stage owing to its anatomical positioning, which may delay the onset of symptoms and subsequent detection. UrCA is associated with a higher propensity for peritoneal metastasis. Studies have indicated that patients with UrCA are generally younger and exhibit more favorable survival outcomes than those with non-urachal adenocarcinoma, likely due to anatomical differences that facilitate earlier detection and a less aggressive disease progression [114].

In localized settings, upfront RC is the standard treatment. In UrCAs, partial cystectomy with excision of the urachal remnant, including the umbilicus, can be considered to mitigate the risk of relapse. The treatment approach for both urachal and non-urachal adenocarcinomas focuses on surgical resection, with wide excision margins to reduce the risk of local recurrence and ensure the complete removal of microscopic tumor deposits [115].

##### Challenges in Perioperative Therapy

There is no established consensus regarding the use of perioperative chemotherapy or chemoradiotherapy for localized BAC or UrCA, primarily because of the lack of high-quality evidence and divergent expert opinions [116]. Ongoing investigations are evaluating the potential benefits of perioperative therapies; however, conclusive guidelines have yet to be formulated.

Chemotherapy regimens commonly employed for colorectal adenocarcinomas, such as platinum-based combinations with 5-fluorouracil (5-FU), folinic acid, and oxaliplatin (e.g., FOLFOX or CAPOX), have demonstrated efficacy in treating advanced BAC and UrCA [117]. This therapeutic efficacy is likely due to shared molecular targets, including overlapping mutations in key oncogenes and tumor suppressor genes such as TP53, KRAS, and APC, as well as eGFR amplification [118].

The response of BAC and UrCA to chemotherapy varies across studies. In a retrospective study at the MD Anderson Cancer Center, frontline treatment with GemFLP, which includes gemcitabine, 5-FU, and cisplatin, was administered to 28 patients with BAC and 40 patients with UrCA [119]. The overall response rates were 35.7% and 20% for BAC and UrCA, respectively. In another study by Galsky et al. [97], a combination of ifosfamide, paclitaxel, and cisplatin yielded an overall response rate of 36% in a cohort consisting of six patients with UrCA and five patients with BAC.

BAC and UrCA share several characteristics with colorectal adenocarcinoma that may influence treatment decisions. These shared features suggest that therapeutic approaches effective against colorectal adenocarcinoma, such as specific chemotherapy regimens or targeted therapies, could potentially yield similar benefits for BAC and UrCA, providing a basis for treatment strategies in the absence of dedicated clinical trials [120,121]. UrCA is characterized by the cytoplasmic expression of β-catenin [122], and both BAC and UrCA exhibit genetic alterations, such as KRAS, MYC, FLT3, and TERT mutations. Notably, UrCAs tend to exhibit a lower frequency of TERT and RB1 alterations than BAC, whereas SMAD4 and GNAS mutations, which are frequently observed in colorectal cancer, are more commonly identified in UrCA [123,124,125]. These molecular features suggest that advanced BAC may benefit from treatment strategies similar to those used for colorectal cancer. The shared molecular profiles provide a rationale for employing similar systemic therapies, which may improve outcomes in patients with advanced disease [121,126].

Immunotherapy, targeted therapy, and ADCs show the potential to manage BAC and UrCA, with early-phase clinical trials suggesting promising new treatment options [3,116]. However, although these therapies are under active investigation, more data are required to confirm their effectiveness, especially given the limited current evidence on ICIs in these rare tumor types. These tumors generally demonstrate a low tumor mutation burden and reduced PD-L1 expression, which are factors associated with less favorable responses to immunotherapy [124,127]. In a small series, patients treated with ICIs demonstrated heterogeneous responses, with reported response rates of 20% for BAC and 25% for advanced UrCA [63,101]. FGFR3 alterations remain rare in BAC and UrCA and were observed in only 1 of 143 BAC cases in one study [124]. The UNITE retrospective dataset showed no response to EV in pure non-urothelial subtype histology, yet mixed adenocarcinoma subtypes with UC components had high response rates (56–63%) based on the adenocarcinoma proportion [64]. A chemoresistant signet ring BAC expressing nectin-4 also responded to late-line EV [128]. While ADCs, such as EV, show limited efficacy in pure adenocarcinoma, higher response rates in mixed adenocarcinomas with UC elements highlight the importance of molecular and histological profiling for tailored treatment.

BAC is a rare and heterogeneous malignancy that shares significant clinical and molecular similarities with colorectal cancer. The scarcity of prospective clinical trial data has necessitated the adoption of new treatment regimens for colorectal cancer, particularly for advanced cases. The evolving landscape of targeted therapies, immunotherapies, and ADCs holds promise for expanding therapeutic options for this challenging subtype of bladder cancer.

The aforementioned content is comprehensively summarized in Table 3.

## 4. Challenges and Future Directions

Histological subtypes of bladder cancer present significant challenges in both diagnosis and management, necessitating specialized research and tailored clinical approaches (Table 4). These subtypes are often characterized by aggressive biological behaviors, poor prognoses, and limited responsiveness to conventional therapies. Furthermore, they frequently manifest in younger patients and exhibit rapid disease progression, often driven by specific molecular markers such as TP53 mutations, FGFR3 alterations, and ERBB2 amplifications [129]. These genetic features contribute to their aggressive nature and highlight the urgent need for refined diagnostic tools and improved therapeutic strategies.

One of the most critical challenges in addressing subtype histology is achieving an accurate diagnosis. These tumors often display overlapping histopathological features, which increase their potential for misclassification. A central pathology review has emerged as an essential mechanism for enhancing diagnostic precision. Studies such as the ABACUS-2 trial have highlighted the impact of a central review, demonstrating that approximately 30% of tumor subtypes are reclassified upon centralized reevaluation [102], which has direct implications for treatment selection and outcomes. For instance, initial classifications, such as SqD, may be revised to plasmacytoid or sarcomatoid upon central review. Such reclassifications are crucial, as they fundamentally affect the understanding of tumor biology and guide appropriate therapeutic approaches. Therefore, enhanced diagnostic accuracy through centralized pathology reviews, along with the essential consideration of advanced tools such as artificial intelligence in digital pathology, is crucial for guiding effective treatment strategies and optimizing patient outcomes [130].

The variability in treatment responses among different subtypes further complicates effective management. Certain subtypes respond differently to chemotherapy, immunotherapy, or targeted therapies. For example, the ABACUS-2 trial revealed that sarcomatoid tumors exhibited a significantly higher pathological response rate to immunotherapy (62%) than SCCs (8%) [131]. Similarly, findings from the UNITED study indicated that the efficacy of EV varied depending on whether the subtype was predominant or a minor component, with lower response rates observed in cases in which the subtype was predominant [132]. These data suggest that the biological behavior of specific subtypes significantly influences therapeutic outcomes, likely due to distinct signaling pathways and genetic mutations such as alterations in the TP53, FGFR3, or PI3K/AKT/mTOR pathways, which affect treatment responses [133]. This highlights the need for subtype-specific treatment protocols.

ADCs and FGFR inhibitors have emerged as promising therapeutic options for urothelial and non-urothelial subtypes. However, as described above, in urothelial and non-urothelial subtypes of bladder cancer, evidence evaluating the efficacy of ADCs and FGFR inhibitors remains limited, unstandardized, and generally associated with lower response rates compared to pure UC. These challenges are compounded by tumor heterogeneity, variability in antigen expression, and discrepancies in FGFR3 mutational status. To advance therapeutic strategies, large-scale prospective clinical trials, detailed molecular profiling, and the adoption of precision-targeted approaches are crucial. Ongoing research is investigating combination therapies—such as FGFR inhibitors paired with ADCs or immune checkpoint inhibitors—to improve outcomes [7].

There is an ongoing debate regarding the inclusion of patients with subtypes in general UC trials versus specialized clinical trials [3]. Including patients with subtypes in broader clinical trials may dilute the results and fail to yield meaningful data given the unique biology of these subtypes and the small sample sizes typically involved. Instead, there is a compelling argument to conduct dedicated international trials that specifically focus on these subtypes. Conducting these specialized trials presents logistical challenges, such as the need for multicenter collaboration, sufficient funding, and patient recruitment. However, the benefits include generating more robust and meaningful data tailored to these unique subtypes, which could significantly improve patient outcomes. Such specialized trials would allow a more accurate assessment of therapeutic efficacy and facilitate the development of targeted therapies that address the specific needs of these patients.

We are rapidly traversing the era of the molecular characterization of bladder cancer across both non-muscle-invasive and muscle-invasive diseases [134]. In the near future, molecular subtyping classifications will likely guide the assessment of prognoses and, more importantly, responses to personalized therapies, ushering in a new treatment paradigm for UC. As we advance, there is a need for an integrated classification system that combines molecular subtyping with traditional histology-based approaches. This integrative framework could offer a more comprehensive understanding of tumor biology, improve predictive accuracy, and enhance personalized treatment strategies for patients with bladder cancer.

Addressing the unmet needs associated with the BC subtype histology of bladder cancer requires global collaboration. Therefore, there is a pressing need for the international genitourinary oncology community to design and implement clinical trials that specifically target these subtypes. Collaborative international efforts can facilitate the accrual of sufficient patient numbers, thereby enabling adequately powered studies to yield robust and generalizable data. Moreover, partnerships with industrial stakeholders and securing appropriate funding are crucial for advancing research initiatives. The ultimate goal is to establish targeted evidence-based treatment protocols that can substantially improve the outcomes of patients with these aggressive cancer subtypes.

## 5. Conclusions

Although uncommon, the subtype histologies of bladder cancer pose a significant challenge for clinicians because of the paucity of prospective data. Patients with these histologic subtypes are frequently excluded from clinical trials focused on UC because of their unique biological characteristics and limited representation, which complicate standardized treatment protocols. This exclusion results in a dearth of robust evidence-based guidelines for their management and affects the generalizability of trial outcomes to these patient populations. Nevertheless, novel therapeutic agents such as enfortumab, vedotin, and pembrolizumab have begun to reshape the treatment paradigm, particularly in patients with advanced or metastatic UC that has progressed after platinum-based chemotherapy, establishing themselves as frontline standards of care. Despite these advancements, it remains imperative to deepen our understanding of the biology and clinical outcomes associated with individual tumor subtypes, rather than relying solely on pooled analyses of heterogeneous histologies, to optimize patient outcomes.

The advent of pan-cancer approvals predicted on target expression marks a promising approach for addressing bladder cancer subtype histologies. For instance, the recent regulatory approval of trastuzumab deruxtecan exemplifies the potential of targeted therapies to provide more effective and biologically tailored treatment options for patients with these rare subtypes [135]. This emerging trend underscores the growing importance of personalized medicine, in which traditional histopathological assessments are increasingly complemented by genotyping and transcriptional profiling to inform systemic therapeutic choices. Advances in technology and bioinformatics have facilitated the identification of molecular signatures and signaling pathways in urothelial cancer, paving the way for precision-targeted therapies and immune-based strategies.

Future clinical trials should be meticulously designed to incorporate this growing body of molecular knowledge. A careful selection of biomarkers should be leveraged to identify actionable therapeutic targets and stratify patients based on predictive and prognostic factors. It is critical to allocate more research resources to elucidate the molecular pathology of rare bladder cancer subtypes that remain relatively understudied. A comprehensive mechanistic understanding of the evolutionary trajectories of bladder cancer through different treatment modalities is essential for the rational combination of chemotherapy, immunotherapy, and other targeted agents, ultimately advancing patient care.

The shift towards a personalized, patient-centric approach to bladder cancer management represents a significant advancement in this field. This paradigm emphasizes precise pathological characterization and rigorous clinical research aimed at developing effective treatment strategies that not only improve clinical outcomes but also enhance patients’ quality of life. By integrating these principles into routine clinical practice, healthcare providers can ensure that therapeutic regimens are tailored to align with the individual preferences and needs of patients, ultimately delivering optimal outcomes for those afflicted with these rare and aggressive subtypes of bladder cancer.

**Table 2 biomedicines-13-00086-t002:** Etiology of urothelial carcinoma subtypes.

Factors	Mechanism/Description	Examples/Effects	Impact on Subtypes	Specific Subtypes Affected	**References**
Genetic Mutations	- Key Gene Alterations: Mutations in TP53, FGFR3, and RB1 disrupt cell cycle control, promote unchecked proliferation, and induce resistance to apoptosis. - Clonal Evolution: Continuous genetic changes lead to multiple genomic clones within tumor.	- Subtype Associations: - FGFR3 Mutations: Common in nested subtypes. - TP53 and RB1 Mutations: Prevalent in sarcomatoid and neuroendocrine subtypes. - Phenotypic Diversity: Contributes to morphological and functional heterogeneity in UC.	Contributes to morphological and functional heterogeneity	FGFR3: Nested subtypeTP53, RB1: Sarcomatoid and neuroendocrine subtypes	[18,19,42,43,44]
Epigenetic Modifications	- DNA Methylation: Hypermethylation of tumor suppressor genes leads to their silencing. - Histone Modifications: Alterations in histone acetylation affect chromatin structure and gene transcription. - Dynamic Changes: Influenced by chronic inflammation, carcinogens, and therapeutic regimens.	- Gene Expression Regulation: Facilitates tumor growth and emergence of subtype histologies. - Adaptive Resistance: Chemotherapy-induced DNA demethylation and histone modifications promote aggressive subtypes. - Differentiation Pathways: Fosters diverse cellular differentiation.	Promotes various differentiation pathways contributing to subtype formation	Affects all subtypes	[20,21,22,23,24]
Tumor Microenvironment	- Cellular Components: Immune cells, fibroblasts, cytokines (IL-6, TNF-α, TGF-β), and extracellular matrix. - Signaling Interactions: Cytokines and hypoxic conditions drive subtype differentiation. - Supportive Niche: Cancer-associated fibroblasts (CAFs) and ECM remodeling facilitate tumor progression.	- Cytokine Effects: - IL-6: Enhances inflammation and immune evasion. - TNF-α: Promotes cellular proliferation and differentiation. - TGF-β: Induces EMT and suppresses immune responses. - Chronic Inflammation: Increases likelihood of squamous differentiation. - Hypoxia: Activates HIFs, driving angiogenesis and metabolic adaptation.	Increases squamous differentiation due to chronic inflammation.Enhances invasiveness and subtype diversity.	Squamous differentiation subtype	[25,26,27,28,29,30]
Cancer Stem Cells	- Plasticity: Ability to shift differentiation states. - Pluripotency Markers: Expression of OCT4, SOX2, and NANOG. - Self-Renewal: Maintains tumor’s aggressive behavior and adaptability.	- Treatment Resistance: CSCs evade chemotherapy and radiation, contributing to recurrence and progression. - Histologic Diversity: CSC plasticity leads to varied histologic subtypes. - Therapeutic Targeting Challenges: Adaptive capacity complicates effective targeting with standard therapies.	Increases therapy resistance and morphological diversity of subtypes	Treatment-resistant subtypes	[33,34,35,36]
Therapeutic Interventions	- Selective Pressure: Treatments apply selective pressures that favor resistant clones. - Genetic and Epigenetic Adaptations: Induced by chemotherapy and radiation. - Immune Modulation: Immunotherapies reshape immune landscape, potentially enabling immune escape.	- Chemotherapy: Can induce sarcomatoid differentiation by selecting resistant clones. - Radiation Therapy: Promotes EMT, increasing invasiveness and subtype morphologies. - Immunotherapy: May influence subtype histology by altering immune interactions and enabling immune escape mechanisms.	Changes in subtypes and increased invasiveness due to therapy	Sarcomatoid subtype and EMT-related subtypes	[37,38,39,40,41]
Association of Specific Genetic Mutations and Subtypes	- Subtype-Specific Mutations: Certain genetic mutations are strongly associated with specific histologic subtypes in UC. - Molecular Profiling: Identification of these mutations through molecular techniques aids in subtype classification and prognosis. - Functional Impact: Mutations drive distinct morphological and functional characteristics of each subtype.	- FGFR3 Mutations: Frequently observed in nested subtypes, contributing to their unique architectural patterns. - TP53 Mutations: Predominant in sarcomatoid and neuroendocrine subtypes, associated with high-grade features and aggressive behavior. - RB1 Mutations: Also prevalent in sarcomatoid and neuroendocrine subtypes, enhancing resistance to apoptosis and promoting metastatic potential. - Clinical Implications: Helps in prognostication and guiding personalized therapeutic strategies based on genetic landscape of UC subtype.	Identifies subtype-specific therapeutic targets and predicts prognosis	Nested, sarcomatoid, and neuroendocrine subtypes	[42,43,44]

Abbreviations: ECM, extracellular matrix; EMT, epithelial–mesenchymal transition; CSC, cancer stem cell.

**Table 3 biomedicines-13-00086-t003:** Summary of urothelial and non-urothelial subtypes in bladder cancer.

Subtype	Incidence	Histological Features	Genetic Alterations	Clinical Features	Treatment Strategies	Prognosis	Response to Therapies	**Unique Challenges**	**References**
Micropapillary Urothelial Carcinoma (MPUC)	0.01–2.2% of urothelial tumors	- Infiltrative tight clusters of micropapillary aggregates - Often associated with vascular and lymphatic invasion - Similar to ovarian papillary serous tumors - Presents with small nests of tumor cells within lacunae	- HER2 overexpression - Luminal and p53-like subtypes identified - Mutations in TP53, FGFR3, and RB1	- Highly aggressive - Often presents at an advanced stage - High metastatic potential - Extensive lymphovascular invasion	- Surgery: Upfront radical cystectomy often recommended - Chemotherapy: Controversial efficacy; retrospective studies show mixed results - Targeted Therapy: HER2-directed therapies (e.g., trastuzumab) - Immunotherapy: Immune checkpoint inhibitors show some efficacy - ADCs: Enfortumab vedotin shows limited efficacy in pure MPUC but better results in mixed histologies	Poor due to aggressive nature and high metastatic potential	- Neoadjuvant Chemotherapy: Conflicting evidence regarding survival benefit - Immunotherapy: Response rates around 28% with checkpoint inhibitors - ADCs: Limited efficacy in pure MPUC; better in mixed subtypes	- Rarity leads to limited prospective clinical trial data - Diagnostic challenges due to debated histopathological criteria - Need for individualized treatment strategies based on disease stage and molecular characteristics	[10,46,47,48,49,50,51,52,53,54,55,56,57,58,59,60,61,62,63,64]
Plasmacytoid Urothelial Carcinoma (PUC)	1–3% of urothelial carcinomas	- Cells resemble plasma cells with high proliferative index - Extensive bladder wall involvement - Invasion of adjacent organs (e.g., ureters, rectum) - Diffuse infiltration along pelvic fascial planes - Similar to linitis plastica phenotype	- CDH1 Alterations: Mutations or promoter hypermethylation leading to loss of E-cadherin - Alterations in p53, RB, DNA damage repair pathways - High tumor mutation burden and immune-infiltrated phenotype	- Presents at muscle-invasive and advanced stages - Early peritoneal dissemination - Symptoms of gastrointestinal and urinary obstruction - Higher-stage disease with positive lymph nodes and surgical margins compared to conventional UC	- Surgery: Upfront radical cystectomy often performed - Chemotherapy: Neoadjuvant cisplatin-based chemotherapy recommended - Immunotherapy: Immune checkpoint inhibitors show response rates of 17–32% - ADCs: Enfortumab vedotin and sacituzumab govitecan show promise in mixed histologies	Very poor, with median overall survival of 19 months	- Neoadjuvant Chemotherapy: Inconsistent response rates - Immunotherapy: Moderate response rates in retrospective cohorts - ADCs: Higher response rates in mixed histologies with UC components	- Diagnostic challenges due to morphological resemblance to plasma cells - Limited prospective clinical trial data - Aggressive nature complicates surgical management - Need for novel therapeutic approaches	[65,66,67,68,69,70,71,72,73,74,75]
Sarcomatoid Urothelial Carcinoma (SUC)	0.1–0.3% of bladder tumors	- High-grade sarcomatous differentiation - Malignant spindle cells with mesenchymal phenotype - Possible differentiation into osteosarcoma, rhabdomyosarcoma, or chondrosarcoma - Mixed epithelial and mesenchymal components	- TP53, RB1, PIK3CA mutations - EGFR or FGFR alterations in basal and double-negative subtypes - High PD-L1 expression	- Highly aggressive - Approximately 50% present with metastatic disease at diagnosis - Poor prognosis	- Surgery: Radical cystectomy recommended, especially for localized disease - Chemotherapy: Neoadjuvant chemotherapy shows survival benefits; platinum-based regimens (e.g., gemcitabine and cisplatin) - Immunotherapy: Immune checkpoint inhibitors (e.g., anti-PD-1/PD-L1) show response rates of 35–40% - ADCs: Enfortumab vedotin and other ADCs under investigation	Extremely poor due to high aggressiveness and metastatic potential	- Neoadjuvant Chemotherapy: Recent meta-analyses show significant survival benefits - Immunotherapy: Promising response rates with checkpoint inhibitors - ADCs: Limited efficacy in pure SUC; ongoing trials evaluating combinations	- Heterogeneous histology complicates diagnosis - Limited treatment options - High levels of PD-L1 expression offer potential but require more research - Need for molecular profiling to guide targeted therapies	[76,77,78,79,80,81,82,83,84,85,86]
Squamous Cell Bladder Cancer (SCC)	2–3% in Western countries - Higher in regions endemic with *Schistosoma haematobium*	- Cohesive tumor cells with keratinization and intercellular bridges - Originates from keratinizing squamous metaplasia - Similar to linitis plastica phenotype	- Loss of chromosome arm 3p - NOTCH1, TP63, and SOX2 mutations	- More prevalent in African regions due to *Schistosoma* infection - High propensity for local invasion - Aggressive local disease - Poor prognosis	- Surgery: Primary treatment is surgical resection (radical cystectomy) - Chemoradiotherapy: Considered when surgery is contraindicated - Chemotherapy: Platinum-based regimens show some efficacy - Immunotherapy: Immune checkpoint inhibitors show response rates of 28% in UC with SqD - ADCs: Enfortumab vedotin and sacituzumab govitecan show varying response rates depending on histologic components	Poor prognosis due to aggressive local invasion	- Chemotherapy: Variable efficacy; some response with platinum-based and taxane regimens - Immunotherapy: Moderate response rates - ADCs: Higher response rates in mixed histologies; limited efficacy in pure SCC	- Rarity leads to limited prospective clinical trial data - Diagnostic challenges due to morphological similarities with plasma cells and other SCCs - Variable response to therapies based on histologic composition - Need for targeted therapies specific to molecular drivers	[87,88,89,90,91,92,93,94,95,96,97,98,99,100,101,102,103,104]
Neuroendocrine Bladder Cancer (Small-Cell Carcinoma)	~2% of bladder cancers - Small-cell carcinoma: 0.5–1.2%	- Small, round, hyperchromatic cells with scant cytoplasm - Immunoreactive for neuroendocrine markers (chromogranin, synaptophysin, CD56, neuron-specific enolase) - Often mixed with non-small-cell carcinoma components - High somatic mutational burden driven by APOBEC-mediated mutations	- TP53, RB1, and TERT promoter mutations - High mutation burden - Loss of TP53 and RB1 contributing to unchecked proliferation	- Rare and highly aggressive - Often presents with metastatic disease (>60% at diagnosis) - Median OS ~12 months for metastatic, <20 months for localized disease	- Surgery: Radical cystectomy for localized disease - Chemotherapy: Platinum-based regimens (etoposide and cisplatin) standard for metastatic disease - Immunotherapy: Mixed results; some trials show response, others do not - Targeted Therapy: DLL3-targeted therapies under investigation (e.g., HPN328) - ADCs: Limited efficacy; nectin-4 low expression	Very poor because of high aggressiveness and early metastasis	- Chemotherapy: Standard platinum-based regimens show initial response but short duration - Immunotherapy: Inconsistent efficacy - Targeted Therapies: Emerging options like DLL3-targeted constructs show promise	- Limited prospective data due to rarity - High metastatic potential complicates treatment - Variable responses to existing therapies - Need for novel targeted and combination therapies	[3,105,106,107,108,109,110,111]
Adenocarcinoma (Non-Urachal and Urachal [UrCA] Subtypes)	~0.8% of bladder cancers UrCA: ~10% of BAC	- Non-Urachal: Various morphological patterns (papillary, nodular, flat, and ulcerative) - Well to moderately differentiated glandular morphology - Resembles colorectal adenocarcinoma - UrCA: Localizes to dome and anterior bladder wall - Enteric, mucinous, signet ring cell, and mixed subtypes	- Non-Urachal: KRAS, MYC, FLT3, and TERT mutations - UrCA: SMAD4, GNAS mutations; lower frequency of TERT and RB1 than that of non-urachal - Shared mutations with colorectal cancer (e.g., TP53, KRAS, and APC)	- Non-Urachal: Prognosis is worse than that ofUrCA - UrCA: Presents at later stages because of anatomical position - Higher propensity for peritoneal metastasis - Typically presents at muscle-invasive and advanced stages - Younger patients with UrCA have better survival outcomes than those with non-urachal	- Surgery: Radical cystectomy for localized disease - Chemotherapy: Platinum-based regimens (e.g., FOLFOX, CAPOX) similar to colorectal cancer - Targeted Therapy: Potential use of therapies effective in colorectal cancer (e.g., EGFR inhibitors) - Immunotherapy: Limited efficacy due to low tumor mutation burden and reduced PD-L1 expression - ADCs: Enfortumab vedotin shows limited efficacy in pure adenocarcinoma but works better in mixed histologies	Generally poor, especially for non-urachal subtypes UrCA has relatively better outcomes due to earlier detection and less aggressive progression	- Chemotherapy: Variable response rates; some efficacy with colorectal-like regimens - Immunotherapy: Heterogeneous responses, generally lower efficacy due to molecular characteristics - ADCs: Better responses in mixed histologies with UC components - Targeted Therapies: Potential benefits from colorectal cancer-targeted approaches	- Extremely rare, limiting prospective clinical trial data - Diagnostic challenges due to morphological similarities with colorectal adenocarcinoma - Need for molecular profiling to guide treatment - Variability in treatment responses based on histologic and molecular subtypes	[112,113,114,115,116,117,118,119,120,121,122,123,124,125,126,127,128]

Abbreviations: MPUC, micropapillary urothelial carcinoma; ADC, antibody–drug conjugate; HER2, human epidermal growth factor receptor 2; SqD, squamous differentiation; APOBEC, Apolipoprotein B mRNA Editing Enzyme, Catalytic Polypeptide-like; OS, overall survival; DLL3, delta-like ligand 3; BAC, bladder adenocarcinoma; EGFR, epidermal growth factor receptor; FGFR, fibroblast growth factor receptor

**Table 4 biomedicines-13-00086-t004:** Key challenges and future directions related to histologic subtypes of bladder cancer.

Category	Details	Examples/References	Future Directions
Diagnostic Challenges	- Increased risk of misclassification due to overlapping histopathological features. - Difficulty in achieving accurate diagnoses.	Approximately 30% of tumor subtypes reclassified in ABACUS-2 trial [102]	- Emphasize importance of central pathology reviews - Utilize advanced tools such as artificial intelligence in digital pathology [130]
Variability in Treatment Response	- Different subtypes respond uniquely to chemotherapy, immunotherapy, and targeted therapies. - Biological behavior influences therapeutic outcomes.	Sarcomatoid tumors: 62% pathological response to immunotherapy vs. squamous cell carcinomas: 8% [131]	- Develop subtype-specific treatment protocols - Establish treatment strategies based on distinct signaling pathways and genetic mutations [133]
Need for Specialized Clinical Trials	- Limitations of General Trials: Including subtype patients in general urothelial carcinoma trials may dilute results and fail to provide meaningful data. - Advantages of Specialized Trials: Dedicated international trials focusing on subtypes can generate robust and meaningful data to improve patient outcomes. - Implementation Challenges: While there are logistical challenges such as multicenter collaboration, funding, and patient recruitment, the benefits outweigh the difficulties.	“UNITED study” showing variable response rates to enfortumab vedotin based on subtype predominance [132]	- Conduct dedicated international trials focusing on specific subtypes - Ensure multicenter collaboration and adequate funding for robust studies
Need for Global Collaboration	- Need for International Efforts: Addressing unmet needs associated with subtypes requires collaboration from international genitourinary oncology community. - Design of Specialized Clinical Trials: It is important to design and implement clinical trials specifically targeting these subtypes. - Industry Partnerships and Funding: Partnerships with industry and securing adequate funding are essential for advancing research initiatives. - Ultimate Goal: These efforts aim to establish evidence-based targeted treatment protocols for aggressive cancer subtypes, significantly improving patient outcomes.		- Strengthen collaboration within international genitourinary oncology community - Build partnerships with industry and secure appropriate funding to advance research initiatives
Future Directions	- Advancements in Molecular Characterization: Rapid progress in molecular characterization of bladder cancer is expected to utilize molecular subtyping for prognosis assessment and personalized therapies. - Integrated Classification System: Need for system combining molecular subtyping with traditional histology-based approaches. - Enhance predictive accuracy and personalized treatment strategies: Such an integrated framework can lead to more accurate predictions and enhanced personalized treatment strategies.		

## Figures and Tables

**Figure 1 biomedicines-13-00086-f001:**
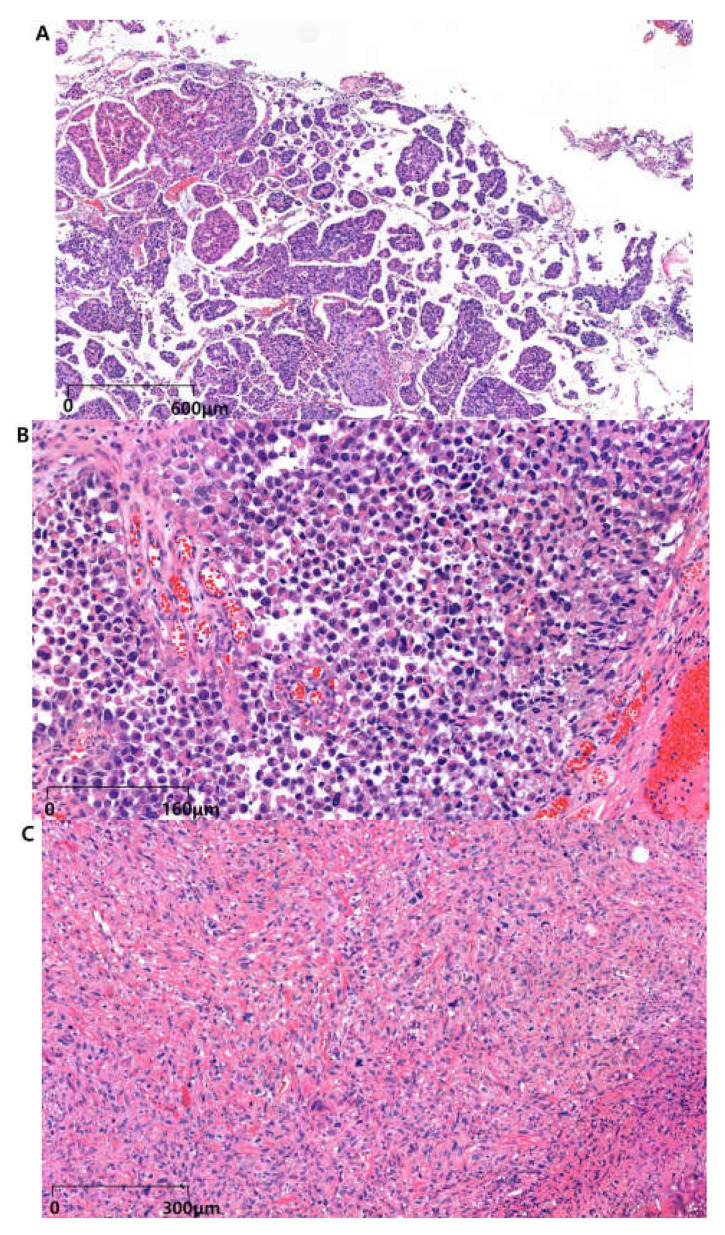
(**A**) Micropapillary subtype of UC (H&E); multiple small nests of tumor cells with surrounding lacunar (empty) space are a classic feature and may be the most helpful feature in making the diagnosis. (**B**) Plasmacytoid subtype of UC (H&E); discohesive single cells with eccentrically placed nuclei and abundant eosinophilic cytoplasm, which are often deeply infiltrative but with minimal stromal reaction. (**C**) Sarcomatoid urothelial carcinoma (H&E); malignant spindled cells with a nonspecific morphologic appearance and a mesenchymal-like growth pattern. The most common component is an undifferentiated high-grade spindle cell sarcoma, as in this figure. (**D**) Bladder squamous cell carcinoma (H&E); keratinization and intercellular bridges, features consistent with squamous differentiation. (**E**) Neuroendocrine bladder cancer (H&E); small, round cells with a high nuclear-to-cytoplasmic ratio, hyperchromatic nuclei, and fine chromatin, typical of neuroendocrine differentiation. (**F**) Bladder adenocarcinoma (H&E); prominent glandular formation of columnar cells and potential mucin production, raising the differential diagnosis of spread from the gastrointestinal tract or other primary sites. Abbreviations: UC, urothelial carcinoma; H&E, hematoxylin and eosin.

**Table 1 biomedicines-13-00086-t001:** The 2022 WHO classification of invasive tumors of the urothelial tract.

Main Histological Subtypes	Subgroups
Invasive UC subtypes	Conventional UC
	Infiltrating UC with squamous differentiation
	Infiltrating UC with glandular differentiation
	Infiltrating UC with trophoblastic differentiation
	Nested UC
	Tubular and microcystic UC
	Micropapillary UC
	Lymphoepithelioma-like UC
	Plasmacytoid UC
	Sarcomatoid UC
	Giant cell UC
	Lipid-rich UC
	Clear cell UC
	Poorly differentiated UC
Squamous cell neoplasms	Pure squamous cell carcinoma
	Verrucous carcinoma
	Squamous cell papilloma
Glandular neoplasms	Adenocarcinoma, NOS–enteric
	Adenocarcinoma, NOS–mucinous
	Adenocarcinoma, NOS–mixed
	Villous adenoma
Tumors of Müllerian type	Clear cell carcinoma
	Endometrioid carcinoma
Neuroendocrine tumors	Small-cell neuroendocrine carcinoma
	Large-cell neuroendocrine carcinoma
	Well-differentiated neuroendocrine tumor
	Paraganglioma
Mesenchymal tumors	Rhabdomyosarcoma
	Leiomyosarcoma
	Angiosarcoma
	Malignant inflammatory myofibroblastic tumor
	Malignant perivascular epithelioid cell tumor
	Malignant solitary fibrous tumor
Miscellaneous tumors	Epithelial tumors of the upper urinary tract
	Tumors arising in a bladder diverticulum
	Urothelial tumors of the urethra
	Malignant melanoma
	Carcinoma of Skene, Cowper, and Littre glands
	Metastatic tumors and tumors extending from other organs

Abbreviations: WHO, World Health Organization; UC, urothelial carcinoma; NOS, not otherwise specified.

## Data Availability

No new data were created.

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
