# Peer review of "Advances in Therapy for Urothelial and Non-Urothelial Subtype Histologies of Advanced Bladder Cancer: From Etiology to Current Development"

_biomedicines, 2025, doi:10.3390/biomedicines13010086_

Round 1
Reviewer 1 Report
Comments and Suggestions for Authors
The manuscript provides a detailed overview of the current classification, management, and emerging therapeutic approaches for bladder cancer subtypes. The authors effectively highlight the challenges associated with the treatment of rare histological subtypes and emphasize the importance of precision medicine. They underline the lack of robust evidence among these subtypes, lacking of clinical trials that include them. However, there are areas where the manuscript could benefit from further refinement and elaboration to enhance its clarity and impact.
The introduction provides a robust context for the review but does not clearly state the main objectives of the manuscript. Consider explicitly stating the primary aim and scope of the review early in the introduction.
The section on antibody-drug conjugates (ADCs) and FGFR inhibitors is promising but could be expanded with more discussion of clinical outcomes, ongoing trials, and potential limitations.
The use of terms like “subtypes” and “variants” may cause confusion. Consider defining these terms clearly at the outset and using them consistently throughout.
Author Response
Reviewer 1
Comments and Suggestions for Authors
The manuscript provides a detailed overview of the current classification, management, and emerging therapeutic approaches for bladder cancer subtypes. The authors effectively highlight the challenges associated with the treatment of rare histological subtypes and emphasize the importance of precision medicine. They underline the lack of robust evidence among these subtypes, lacking of clinical trials that include them. However, there are areas where the manuscript could benefit from further refinement and elaboration to enhance its clarity and impact.
1- The introduction provides a robust context for the review but does not clearly state the main objectives of the manuscript. Consider explicitly stating the primary aim and scope of the review early in the introduction.
Response: Thank you for your thoughtful suggestion. We added sentences about the primary aim and scope of the review early in the introduction.
Revisions:
Page 2, line 54-58
In this review, we aim to provide a comprehensive synthesis of the current management landscape for advanced bladder cancer—encompassing both urothelial and non-urothelial subtypes—by integrating histopathological insights, evolving therapeutic strategies, and emerging molecular targets to better inform individualized treatment approaches.
2- The section on antibody-drug conjugates (ADCs) and FGFR inhibitors is promising but could be expanded with more discussion of clinical outcomes, ongoing trials, and potential limitations.
Response: Thank you for the valuable suggestion. We have added paragraph to the 4. Challenges and Future Directions section for more discussion of clinical outcomes, ongoing trials, and potential limitations.
Revisions:
Page 17, line 703-712
ADCs and FGFR inhibitors have emerged as promising therapeutic options for urothelial and non-urothelial subtypes. However, as described above, in urothelial and non-urothelial subtypes of bladder cancer, evidence evaluating the efficacy of ADCs and FGFR inhibitors remains limited, unstandardized, and generally associated with lower response rates compared to pure UC. These challenges are compounded by tumor heterogeneity, variability in antigen expression, and discrepancies in FGFR3 mutational status. To advance therapeutic strategies, large-scale prospective clinical trials, detailed molecular profiling, and the adoption of precision-targeted approaches are crucial. Ongoing re-search is investigating combination therapies—such as FGFR inhibitors paired with ADCs or immune checkpoint inhibitors—to improve outcomes [7].
3- The use of terms like “subtypes” and “variants” may cause confusion. Consider defining these terms clearly at the outset and using them consistently throughout.
Response: Thank you for the valuable suggestion. We have added sentences to the introduction to reduce confusion about terms like “subtypes” and “variants” and use “subtypes” consistently throughout. Table 1 was also revised accordingly.
Revisions:
Page 1-2, line 42-50, (minor: 75,113,285,478,655-656) and Table 1
The terminology “Urothelial Subtypes” refers to categories of bladder cancer that originate from urothelial cells but exhibit additional and distinct histological features beyond the conventional UC. These include UCs with mixed histological differentiation, such as squamous or glandular differentiation, or micropapillary UC. In con-trast, “Non-Urothelial Subtypes” describe bladder cancers that arise from entirely different cellular lineages and not from urothelial cells. These subtypes include squamous cell carcinoma (SCC), adenocarcinoma, and small cell carcinoma, which emerge through alternative pathways of differentiation and are distinct in both their histological appearance and cellular origin.
Table 1. 2022 WHO classification of invasive tumors of the urothelial tract
|
Main histological subtypes |
Subgroups |
|
Invasive UC subtypes |
Conventional UC |
|
Infiltrating UC with squamous differentiation |
|
|
Infiltrating UC with glandular differentiation |
|
|
Infiltrating UC with trophoblastic differentiation |
|
|
Nested UC |
|
|
Tubular and microcystic UC |
|
|
Micropapillary UC |
|
|
Lymphoepithelioma-like UC |
|
|
Plasmacytoid UC |
|
|
Sarcomatoid UC |
|
|
Giant cell UC |
|
|
Lipid-rich UC |
|
|
Clear cell UC |
|
|
Poorly differentiated UC |
|
|
Squamous cell neoplasms |
Pure squamous cell carcinoma |
|
Verrucous carcinoma |
|
|
Squamous cell papilloma |
|
|
Glandular neoplasms |
Adenocarcinoma, NOS -Enteric |
|
Adenocarcinoma, NOS -Mucinous |
|
|
Adenocarcinoma, NOS -Mixed |
|
|
Villous adenoma |
|
|
Tumors of Müllerian Type |
Clear cell carcinoma |
|
Endometrioid carcinoma |
|
|
Neuroendocrine tumors |
Small cell neuroendocrine carcinoma |
|
Large cell neuroendocrine carcinoma |
|
|
Well-differentiated neuroendocrine tumor |
|
|
Paraganglioma |
|
|
Mesenchymal tumors |
Rhabdomyosarcoma |
|
Leiomyosarcoma |
|
|
Angiosarcoma |
|
|
Malignant inflammatory myofibroblastic tumor |
|
|
Malignant perivascular epithelioid cell tumor |
|
|
Malignant solitary fibrous tumor |
|
|
Miscellaneous tumors |
Epithelial tumors of the upper urinary tract |
|
Tumors arising in a bladder diverticulum |
|
|
Urothelial tumors of the urethra |
|
|
Malignant melanoma |
|
|
Carcinoma of Skene, Cowper, and Littre glands |
|
|
Metastatic tumors and tumors extending from other organs |
Abbreviations: WHO, World Health Organization; UC, urothelial carcinoma; NOS, not otherwise specified
Reviewer 2 Report
Comments and Suggestions for Authors
This review present recent advances in management of urothelial / non-urothelial bladder cancer subtypes and try to explore the current evidence guiding the treatment and emphasize the challenges and perspectives of future therapeutic strategies. The article is well-documented, logical, and in line with the scope of this journal.
Minor issue should be modified.
In Figure 1, no scale bar was shown in image. The images from different subtype of UC are inconsistent and not aligned, which lacks the rigor of scientific research papers and affects readers' perception.
Author Response
Reviewer 2
This review present recent advances in management of urothelial / non-urothelial bladder cancer subtypes and try to explore the current evidence guiding the treatment and emphasize the challenges and perspectives of future therapeutic strategies. The article is well-documented, logical, and in line with the scope of this journal.
Minor issue should be modified.
In Figure 1, no scale bar was shown in image. The images from different subtype of UC are inconsistent and not aligned, which lacks the rigor of scientific research papers and affects readers' perception.
Response: Thank you for the valuable suggestion.
Revisions: We replaced with an image with a scale bar added. (Figure 1.)